# Ecological Network Construction of a National Park Based on MSPA and MCR Models: An Example of the Proposed National Parks of "Ailaoshan-Wuliangshan" in China

**Caihong Yang [1], Huijun Guo [2,\*], Xiaoyuan Huang [1,\*] 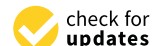, Yanxia Wang [1], Xiaona Li [1] and Xinyuan Cui [1]**

1   School of Geography and Ecotourism, Southwest Forestry University (SWFU), Kunming 650224, China
2   National Plateau Wetlands Research Center, Southwest Forestry University (SWFU), Kunming 650233, China
\*   Correspondence: hjguo@swfu.edu.cn (H.G.); hxy21cn@swfu.edu.cn (X.H.); Tel.: +86-138-8824-1697 (X.H.)

**Abstract:** The establishment of ecological networks facilitates genetic exchange among species in national parks and is an effective means of avoiding habitat fragmentation. Using the proposed "Ailaoshan-Wuliangshan" in Yunnan Province, China, as the study area, the identification of ecological source sites using the morphological spatial pattern analysis (MSPA) method, extraction of potential ecological corridors using the minimum resistance model (MCR) and construction of the ecological network of national parks were performed. Based on the gravity model, important ecological corridors were selected, and corresponding ecological network optimization strategies were presented. The results showed that (1) the core area identified by MSPA was 4440.08 km$^2$, with a low degree of fragmentation, and is distributed in strips within the woodland land classes in the study area; (2) the establishment of an ecological network model of least cost resistance based on 10 indicators in four dimensions of land tenure, geographic factors, vegetation characteristics, and human meddling; (3) the ecological network included 13 ecological source sites, 77 potential ecological corridors, 48 important ecological corridors and 25 pedestrian pathways and extracts an optimal ecological corridor connecting with the natural reserve; and (4) the network closure degree of the constructed ecological network was (1.18), line point rate (3.08), network connectivity (1.12), and cost ratio (0.98). By using the proposed ecological network construction method, ecological patches and potential corridors can be accurately identified to ensure the integrity and connectivity of the national park while minimizing the land demand pressure of the surrounding communities, which provides some reference for the construction of other national parks' ecological networks in China.

**Keywords:** national park; MSPA; MCR; ecological corridor

## 1. Introduction

China is one of the hotspots of biological habitats and biodiversity [1], and several critical biodiversity areas, including the Hengduan Mountains, the Tibetan Plateau mountains, Xishuangbanna in southern Yunnan, the Qinling Mountains, the Changbai Mountains, and the Tianshan Mountains, have been proposed based on species richness and the number of endemic species [2]. With the rate of urbanization accelerating, the natural environment is being destroyed to varying degrees, and landscape connectivity is diminishing. Habitat fragmentation can lead to the isolation of biological populations, which greatly increases the likelihood of extinction and poses a serious threat to biodiversity [3]. The national park system is designed to effectively protect the originality and integrity of the most nationally representative natural ecosystems with a broad scope of protection and comprehensive ecological processes. In June 2019, the General Office of the Central Committee of the Communist Party of China and the General Office of the State Council issued the Guiding Opinions on the Establishment of a Nature Reserve System with National Parks as the Mainstay, emphasizing the main position of the conservation value and ecological functions of national parks in the national nature reserve system. No other types of nature reserves

will be maintained or established in the same areas after national parks are established [4]. Official approved national parks due to the late start of national park research in China are Giant Panda National Park, SanJiangYuan National Park and Northeast Tiger National Park as well as Leopard National Park HaiNan Tropical Rainforest and WuYiShan National Park [5], which have a large gap with the construction of the proposed national park-based nature reserve system in China [6]. Despite the large number of nature reserves of various types and functions, China has played an important role in biodiversity protection and national ecological security maintenance. However, due to the fragmented distribution and fragmentation of nature reserves [7], their variable size and small protection areas, and the distribution of a large number of remaining forests, villages, towns, and agricultural lands around them, ecosystem integrity is blocked, and integrity and connectivity are not robust [8]. Both the Conservation for Biodiversity Aichi Targets and the Post-2020 Global Biodiversity Framework (GBF), which is under discussion globally, emphasize the importance of PA connectivity and set global PA connectivity targets [9]. Aichi Targets and the First Draft of the Post-2020 GBF call for 17% and 30%, respectively, of the global land area to be conserved through well-connected PAs and other effective area-based conservation measures (OECMs) [9,10]. Therefore, how to build an ecological network system and realize ecosystem integrity by relying on existing nature reserves is the key issue facing national park building.

Ecological networks are made up of patches and their connections to achieve effective conservation of species diversity through the establishment of ecological corridors through fragmented natural systems [11]. In terms of construction methods, "identifying ecological sources—constructing resistance surfaces—extracting ecological corridors" has become the basic framework for constructing ecological networks [12,13]. There are usually two methods for determining ecological source sites: one is to directly select nature reserves, attractions, and forest parks as ecological source sites based on the empirical judgement of professionals [14], which is subject to more subjective interference and ignores the connecting role of patches in the landscape [15]. Second, morphological spatial pattern analysis (MSPA), proposed by Vogt et al. to achieve the measurement and identification of spatial patterns of forest landscapes by correlating morphological features with specific shapes in raster images [16], is widely used in forest fragmentation and urban green space system research [17,18]. This method is different from the traditional method of selecting only nature reserves, forest parks, etc., as ecological source sites can classify the spatial pattern of raster images more precisely in terms of functional-type structures and then identify landscape types with different ecological meanings and increase the scientificity of ecological source sites [19]. Species migration and exchange between different ecological source sites can only be achieved if resistance is overcome, and the resistance surface is the total cost of overcoming multiple resistance factors formed between patches during species migration. The ease of species migration between different landscape units varies. The higher the suitability of the patch is, the lower the resistance of species migration, and the resistance is mainly influenced by factors such as topography, land use type and the intensity of human interference. Combining the basis of existing resistance surface-related studies at home and abroad [20–22], this study constructs an ecological network resistance surface based on 10 indicators in four dimensions: land tenure, geographic factors, vegetation characteristics and human interference. Ecological corridors can improve landscape connectivity and contribute to species dispersal and maintain gene flow between populations. Many methods have been used today to identify ecological corridors, such as individual-based movement models, connectivity probability (PC), and circuit theory [23–25]. Currently, the minimal cumulative resistance model (MCR) has become the mainstream method for identifying ecological corridors. The method was first proposed in 1992 by Knaapen et al. by calculating the minimum consumption path between the source and target and the optimal path for the outward spread, migration and dispersal of species [26], which can effectively avoid interference from the external environment and well reflect the possibilities and trends of movement of living species between habitat areas, thus protecting

biodiversity [27–29]. There are an increasing number of studies combining MSPA and MCR models to construct ecological networks, but they are mostly used for urban ecological network construction by identifying the central nodes of urban ecological source sites and establishing urban ecological networks in combination with road and water networks and mainly focusing on the construction of ecological networks within the city (a whole) [30,31]. Currently, there are few studies on the construction of ecological networks in national parks [32,33]. For example, the ecological networks of Giant Panda National Park and Shuangzi Mountain National Forest Park were constructed by using 3S technology and the theory of landscape ecology to identify ecological source sites. The least cumulative resistance method was used to simulate important corridors and potential corridors, and an ecological network optimization strategy was proposed. The ecological network of urban parks based on birds is constructed by using several factors in the InVEST model to determine the suitable ecological source sites for birds, and the corridor is extracted by constructing a resistance surface with three indicators: land use type, road and water system. This study will address the problem of insufficient spatial connectivity in national parks composed of multiple nature reserves and provide a scientific basis for achieving national park connectivity and integrity by constructing potential ecological corridors and ecological networks using MSPA and MCR models. The objectives of this study were as follows: (1) Build an ecological network of national parks with multiple protected areas to improve the integrity and connectivity of national park ecosystems. (2) The identification of ecological corridors and ecological networks in national parks using a combination of MSPA and MCR models. (3) The determination of the resistance surface involves several factors, and the weight of the human interference factor was set to a higher value to form a more reasonable ecological resistance surface.

## 2. Materials and Methods

### 2.1. Study Area

The proposed "Ailaoshan-Wuliangshan" National Park, consisting of the Ailao Mountains National Nature Reserve, WuLiang Mountains National Nature Reserve and Dinosaur River State Nature Reserve, is located in the central part of Yunnan Province within Jingdong County's territory, Zhenyuan County, Xinping County, Chuxiong City, Shuangbai County, Nanhua County, and Nanjian County, which are linked by four states (cities), namely, Pu'er City, Yuxi City, Chien Yi Autonomous Prefecture, and Dali Bai Autonomous Prefecture, with geographic coordinates. The geographical coordinates are 23°46′50.75″~24°56′06.35″ north latitude and 100°19′07.95″~101°37′54.19″ east longitude (Figure 1).

With a length of 180 km from north to south, a width of 130 km from east to west, an elevation of 452~3348 m and a total area of 1652.82 km$^2$. The "Ailaoshan-Wuliangshan" Mountains Conservation Area is a major conservation area and boundary zone in China, located at the intersection of two geographic units, the Hengduan Mountains in western Yunnan and the plateau in eastern Yunnan, and is a significant corridor for tropical to temperate transition, species migration, and gene exchange in the Asian continent, as well as one of eight routes for global migration of birds. Ecological security, with the complex biodiversity composition and obvious transitional characteristics of the flora and fauna, directly affects Vietnam, Laos, Myanmar and even many countries in the southern and southeastern subregion bordering Yunnan [34]. In parallel, the region is also the water catchment area and ecological conservation area of two major cross-border rivers, namely, the Lancang River (Mekong) and Yuanjiang River (Red), which has great significance to maintain the international ecological security of the area. The Mount Ailao Mountains–Wuliang Mountains are a typical representative of the subtropical forest ecosystem and belong to the priority land protection ecosystem of China, among which the Zhongshan broadleaf wet evergreen forest area is the largest and most comprehensive broadleaf leaf wet evergreens in China, preserving the largest area of broadleaf mountain evergreen forests in the subtropics of China, which is a completely primitive state, has stable natural breeding

resources, and has extremely rich wildlife resources [35,36]. In both mountainous areas, the diversity of animals is extremely rich, with over 90% of the western black-crowned gibbon (*Nomascus concolour*) populations inhabiting both mountainous terrains [37]. There are also nationally protected keystone I animals such as the grey langur (*Trachypithecus phayrei*), nationally protected keystone I birds (*Syrmaticus humiae*), and green peafowl *(Pavo muticus)* in the region, and the biodiversity composition is complex [38]. Lamentation "Ailaoshan-Wuliangshan" National Park can more effectively integrate conservation efforts, establish perfect protection mechanisms and maintain the originality and integrity of ecosystems. In this study, the three nature reserves identified by the proposed national park and the area between them are taken as a whole, with a 1.3 km buffer outwards as a study region.

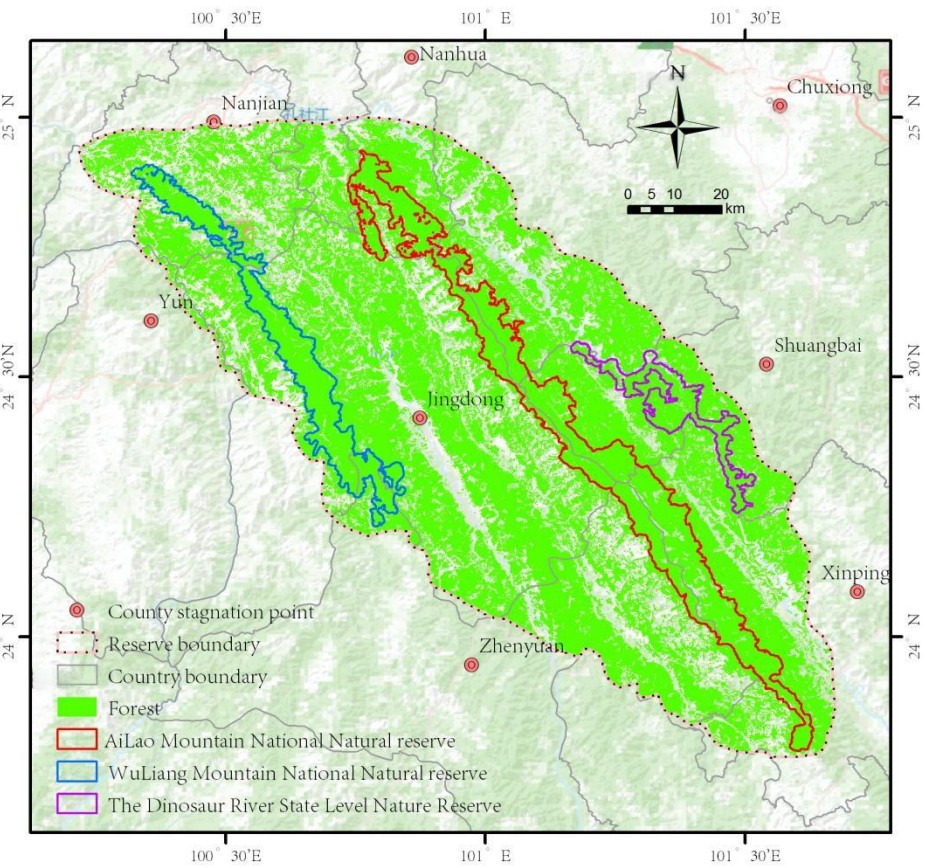

**Figure 1.** Geographic location map of national parks.

### 2.2. Data Sources and Preprocessing

In the study area, vector data were the land use type data of 2020 obtained from the Pu 'er Third and Fourth Forestry Resources Type II Survey database sponsored by the Pu 'er Forestry and Grassland Bureau. This database was created using SPOT-5 satellite imagery at 2.5 m resolution combined with field surveys; vector data are grouped by multiple attributes, such as land use type, land ownership, vegetation origin and forest type. Forest-land categories were used in this study to determine forest distribution, with farmland and built-up areas reflecting anthropogenic disturbance. We obtained township boundaries using BIGEMAP, a Google package that facilitates the editing of satellite maps for upload. Elevation and slope data were generated from a 30 m digital elevation grid (DEM) provided by the Geospatial Data Cloud Platform of the Computer Network Information Center of the Chinese Academy of Sciences "http://www.gscloud.cn (20 September 2021)". The study area of 9914.24 km$^2$ was obtained by buffering outward from the three protected areas as a whole. The land use types in the study area were classified into eight categories, including forest, shrubland, unstocked forest, agricultural land, water bodies, cropland, buildings, and other non-forest land. The habitat characteristics of the western black-crowned gibbon,

the flagship species of the proposed Mourning Mountains–Wuilangshan National Park, have been identified as an important basis for the construction of the resistance surface of the ecological system [39].

*2.3. Methods*

2.3.1. Identifying Ecological Sources Based on MSPA

Morphological spatial pattern analysis (MSPA) is a method for the quantitative identification of ecological source lands and is mainly used to identify and classify ecological source sites by image methods to derive a more scientific distribution of ecological source sites [40]. This study, based on the accurately calibrated land use data of the national park, reclassifies the already classified primary land classes, extracts forestland as the foreground, uses remaining arable land, water, other land, and building land as the background, and converts it into binary tiff maps. The data were analyzed for morphological spatial patterns using Guidos' eight-neighborhood software method to obtain seven landscape element categories that are independent of each other and have different landscape functions, namely, Core, Islet, Bridge, Edge, Perforation, Loop and Branch [41]. Of these, core areas are the largest habitat patches of the seven landscape types, have higher connectivity, are more conducive to species survival and spread and are important for maintaining ecosystem integrity and biodiversity [42]. Lastly, the classification results were tallied, and considering the balanced distribution of ecological patches, 30 ecological source sites were selected for landscape importance analysis based on the area and spatial distribution area of the MSPA core patches.

2.3.2. Evaluating the Importance of Ecologically Sourced Landscapes

Higher importance indices of patches represent more stable ecosystems within the source site. Referring to the relevant literature [43], two landscape indices, *pc* and *dpc*, were selected in this study and calculated using Conefor2.6 software.

This is example 1 of an equation:

$$PC = \sum_{i=1}^{n} \sum_{j=1}^{n} \alpha_i \times \alpha_j \times q_{ij}^* / A, dpc = 100\% \times (pc - pc_{remove}) / pc, \tag{1}$$

where *pc* is the probability connectivity index, *dpc* is the patch importance index, and *A* represents the total area of the landscape. N is the total number of patches, $\alpha_i$ and $\alpha_j$ are the areas of patches *i* and *j*, respectively, $q_{ij}$ is the maximum distance for organisms to spread in different patches, and pcremove is the overall connectivity index of the landscape after removing a patch. The larger the *dpc* value is, the more important the interelement ratio is. Considering that the distance threshold was set too large, which will result in splitting some large patches and vanishing small patches, the patch connectivity distance threshold value was set to 500 m and 0.5 as the connectivity probability between patches [44].

2.3.3. Ecological Network Construction Method

1. Construction of Integrated Resistance Surface

In the proposed Ailao and WuLiang Mountains National Park, species are inevitably hampered by different factors and degrees in the migration process of each source location. Currently, most studies select three resistance factors, namely, land use type, topography, and slope, to construct resistance surfaces [15]. In this study, the resistance surface was constructed by combining land ownership, human disturbance (town center, village, road, land type), vegetation (vegetation type and origin of tree species) and geographic factors (elevation and slope), and each resistance factor was assigned different weight values, with higher weight values indicating a greater influence of the resistance factor on the migration of biological species; in contrast, lower weight values indicated less influence.

Land tenure factor weight, human disturbance factor weight value, vegetation factor weight and geographical factor weight were determined according to related studies [45,46]

(Table 1). The land tenure factor was divided into two types, national and collective, each with a weight value of 0.5 each; the vegetation factor was set to 0.6 and 0.4 strength values for the vegetation type and the origin of tree species according to habits and activity characteristics of the species; the influence of anthropogenic disturbance was set at 0.3, 0.2, and 0.2, respectively, according to the distance of species from town centers, villages, and roads during the migration process. The land use type was set at 0.3 according to the influence of the land type on species; the geographical factors included elevation and slope, with the elevation set at 0.5 according to the characteristics of species' activities; and the slope was set at 0.5 according to the standard grading table of woodland slopes. The weighted overlay operation by the ArcGis matrix calculator was used to build the comprehensive Two Mountains National Park resistance surface as the cost data of the MCR model [46]. The equation is as follows:

**Table 1.** Assignment of resistance factors.

| Resistance Factors | Weight | Classification Indications | Resistance Value |
|---|---|---|---|
| Land ownership | 0.2 | State owned | 0.5 |
| | | Community owned | 0.5 |
| Artificial | 0.4 | Town center | 0.3 |
| | | Village | 0.2 |
| | | Road | 0.2 |
| | | Land use type | 0.3 |
| Vegetation | 0.3 | Type | 0.6 |
| | | Origin | 0.4 |
| Landform | 0.1 | Altitude | 0.5 |
| | | Slope | 0.5 |

This is example 2 of an equation:

$$F_i = \sum_{j=1}^{n} W_j \times A_{ij}, \tag{2}$$

where *i* represents the grid, *j* represents the resistance factor, $F_i$ represents the integrated resistance value of grid *i*, *n* represents the number of resistance factors, $W_j$ represents the proportion of *j* and $A_{ij}$ represents the strength value of *j* in grid *i*.

2. Potential ecological corridor construction based on the MCR model

The minimum cumulative resistance (MCR, minimal cumulative resistance model) model was first introduced into China by Yu Kongjian [47]. It can determine pathways by calculating the minimum cumulative resistance distance between the source and target to better reflect the physical energy of the landscape and the likelihood of biological species moving between habitat patches and trends [27]. The cumulative surface area of minimum resistance for the expansion of ecological source sites in all directions can be obtained by using the model of minimum strength [26].

This is example 3 of an equation:

$$MCR = f_{min} \sum_{j=n}^{i=m} (D_{ij} \times R_i), \tag{3}$$

where *MCR* refers to the minimum cumulative resistance value of the ecological source to one another point; fmin is the minimum cumulative resistance value (*MCR*), representing the positive correlation function; $D_{ij}$ indicates the spatial distance to be crossed for a point *j* to reach another point *i*; and $R_i$ is the resistance value to be overcome across space i.

The cost distance tool in ArcGis distance analysis was used to generate the minimum cumulative resistance surface using ecological source sites and integrated resistance sur-

faces [41], and the cost path tool was used to calculate the minimum cost path from the source site to the target to generate potential ecological corridors.

3.  Determination of the ecological nodes

Ecological nodes are the point of intersection of the pathways and the shortest routes needed by species during migration and are the weakest ecological functions with a "stepping stone" role [48]. For organisms that migrate long distances, increasing the number of "stepping stones" and decreasing the distance between "stepped stones" may effectively improve species survival rates during migration [44]. In combination with the study area environment, the collection points of the minimum cost path and the ecological patches of the bridging area are used as the ecological nodes.

4.  Identify important ecological corridors

The gravity model can scientifically and quantitatively evaluate the strength of interactions between patches, and the larger the value of interaction force is, the more important the position of the corridor between them in the ecosystem of the study area [49–51].

This is example 4 of an equation:

$$G_{ab} = \frac{N_a N_b}{D_{ab}^2} = \frac{L_{max}^2 \ln s_a \ln s_b}{L_{ab}^2 p_a p_b} \tag{4}$$

where $G_{ab}$ is the interaction strength between ecological source sites $a$ and $b$; $N_a$ and $N_b$ represent the corresponding weight values of source sites $a$ and $b$; $D_{ab}$ is the standard value of corridor resistance between source sites; $P_a$ and $P_b$ represent the average resistance values of source sites $a$ and $b$; $S_a$ and $S_b$ are the areas of source sites $a$ and $b$; $L_{ab}$ is the value of corridor resistance between source sites $a$ and $b$; and $L_{max}$ is the minimum cumulative resistance in the area of the maximum value.

According to the construction of potential ecological corridors, the interaction matrix between ecological source sites was computed using a gravity model to quantitatively analyze the strength of interactions among patches. Higher values of interaction force between source patches indicate less resistance and closer contact at ecological sources, the more frequent the material–energy transfer, information transfer, species migration and the more important the corridors connected between them [52]. According to the calculation results of the gravity model and the actual situation in the research area, the interaction strength of the potential ecological corridors greater than 700 is regarded as important corridors and other corridors as general corridors. Finally, the ecological network map of the proposed Ailao–Wuliang Mountains National Park is obtained.

### 2.3.4. Ecological Network Connectivity Evaluation

The graph theory and network analysis method were used to assess the ecological network connectivity of Ailao–Wuliang Mountains National Park and explore the effectiveness of its internal structure. The four factors of network closure ($\alpha$), line point rate ($\beta$), network connectivity ($\gamma$) and cost ratio (c) were used to determine the connectivity of the ecological network in the study area [53].

This is example 5 of an equation:

$$\begin{aligned} \alpha &= (L - v + 1)/(2v - 5) \\ \beta &= \frac{L}{v} \\ \gamma &= \frac{L}{L_{max}} = \frac{L}{3(v-2)} \\ c &= 1 - \frac{L}{d} \end{aligned} \tag{5}$$

where $L$ denotes the number of corridors, v denotes the number of ecological nodes, and d is the total length of all corridors in the ecological corridor. A higher $\alpha$ index indicates a greater number of circuits in the ecological network and greater material circulation and energy mobility [54]. $\beta$ is the number of corridors corresponding to the ecological

nodes, $\beta < 1$ is a tree-like ecological corridor, $\beta = 1$ is a single-loop ecological network, and $\beta > 1$ is a complex ecological network structure. $\gamma$ in [0,1], characterizing the degree of interconnection of ecological nodes in the network, and a larger value of $\gamma$ indicates a higher degree of interconnection of ecological nodes. $c$ indicates the input/output relationship, and a lower value is more favorable for building ecological networks.

## 3. Results and Analysis

### 3.1. Subsection Analysis of Ecological Source Results Based on MSPA

The MSPA was performed with forested land in the study area as the foreground (Figure 2), and the area and proportion of each type of landscape were also counted (Table 2). Among them, the total foreground area is 7820.08 km$^2$, accounting for 79% of the total survey area, which mainly consists of the core and bridge areas. Among the various types of foreground landscapes, the core area is the largest, accounting for 56.78% of the total area, with more large patches in the core region and distributed in bands within the national park, which are not far apart and conducive to the overall connectivity of the region under study. Bridging zones with a larger area of 22.08% indicates that the connectivity between core areas of prospective patches is high, which is conducive to the circulation of organisms between core zones; the edge zone and the pore space both have edge effects and can maintain the stability of the core areas, with proportions of 9.45% and 2.72%, respectively, which indicates that core areas in this study area are relatively stable. In addition, island plaques accounted for the lowest proportion, at percent, indicating that there were few isolated, fragmented and disconnected patches within the study; the proportions of ring roads and spurs were 4.88% and 3.09%, respectively. In general, the large ecological patches in the study area are more concentrated, the landscape connectivity is better and the edges are more stable, which are conducive to the construction and optimization of ecological networks.

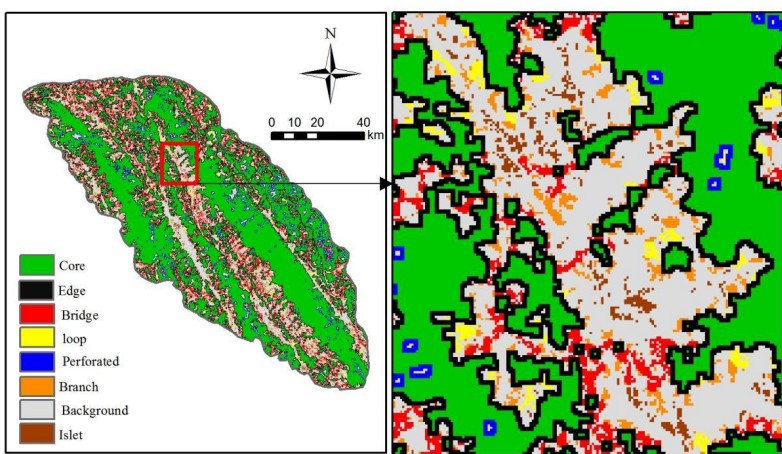

**Figure 2.** Landscape classification map based on MSPA.

**Table 2.** Area of each landscape type based on MSPA.

| Landscape Type | Area (km$^2$) | Proportion of Forest and Areas (%) | Proportion of Total Area (%) |
|---|---|---|---|
| Core | 4440.08 | 56.78 | 44.78 |
| Islet | 77.91 | 1 | 0.79 |
| Perforation | 212.54 | 2.72 | 2.14 |
| Edge | 739.37 | 9.45 | 7.46 |
| Loop | 381.35 | 4.88 | 3.85 |
| Bridge | 1727.05 | 22.08 | 17.42 |
| Branch | 241.79 | 3.09 | 2.44 |
| Total | 7820.08 | 100 | 78.88 |

### 3.2. Analysis of the Importance of Ecological Source Landscapes

Thirteen ecological patches with large values of the importance of patches were selected as habitats for migration and development and reproduction of biological species based on the calculation results of the software Conefor2.6 (Table 3). From Table 3 and Figure 3f, the importance index of patch 9 is 70.9, with an area of 1231.16 km², and it is located in the protection zone of Ailao Mountain, which indicated that the landscape connectivity within this region is good, which is conducive to species migration activity among patches. Figure 4 is followed by patch 3, with an importance index of 17.73 and an area of 750.89 km², which is located within the boundaries of the Wuliang Mountains Nature Reserve. Ecological patches 4 and 11, with a larger remaining area and higher importance index, were also distributed in the nature reserve. We can see from (f) that these patches serve both as habitats for species and corridor connectivity across the landscape. Additionally, other large patches were densely distributed around the reserve, such as 5, 6, 8 and 10. These patches facilitate species migration between reserves and promote connectivity of the overall landscape of the study area.

**Table 3.** Evaluation table of the landscape importance of the ecological source.

| Code | Area/km² | dPC | Code | Area/km² | dPC |
|---|---|---|---|---|---|
| 1 | 31.94 | 0.03 | 8 | 447.37 | 7.00 |
| 2 | 9.39 | 0.00 | 9 | 1231.16 | 70.90 |
| 3 | 750.89 | 17.73 | 10 | 57.09 | 1.41 |
| 4 | 128.18 | 14.98 | 11 | 191.73 | 17.26 |
| 5 | 184.95 | 8.60 | 12 | 20.33 | 0.94 |
| 6 | 33.19 | 1.38 | 13 | 17.64 | 0.23 |
| 7 | 74.59 | 1.78 | | | |

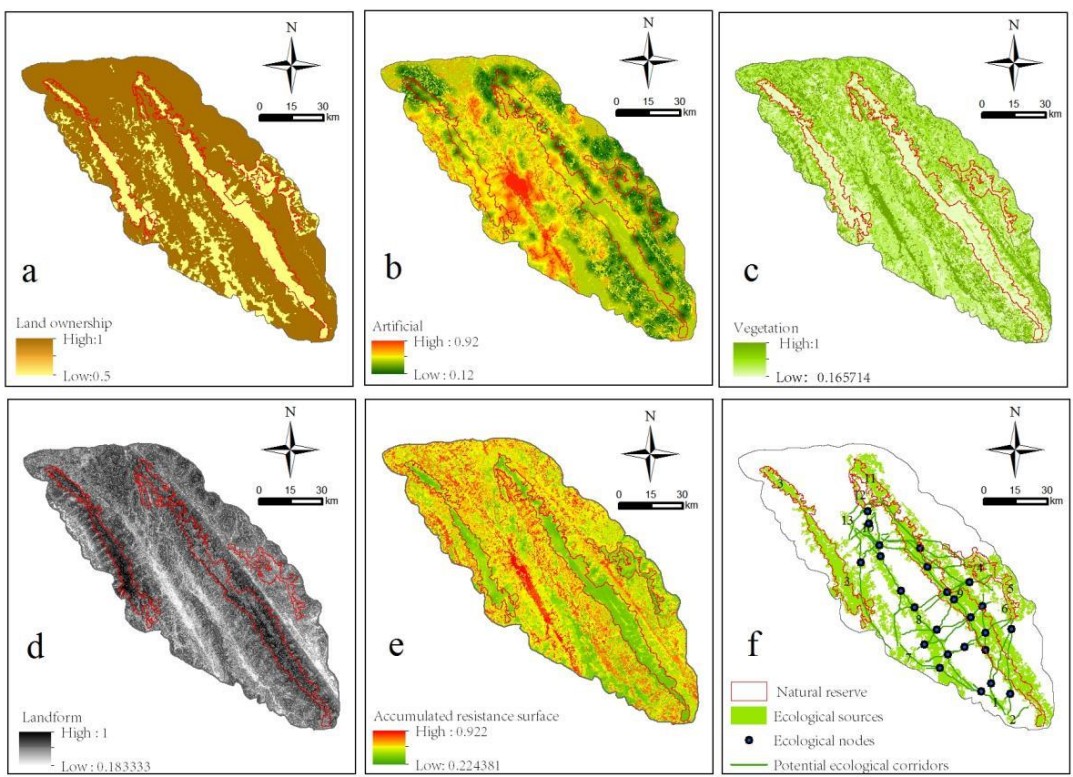

**Figure 3.** Composite resistance surface of species migration in national park. (**a**) Land ownership; (**b**) artificial; (**c**) vegetation; (**d**) landform; (**e**) accumulated resistance surface; and (**f**) potential ecological corridors.

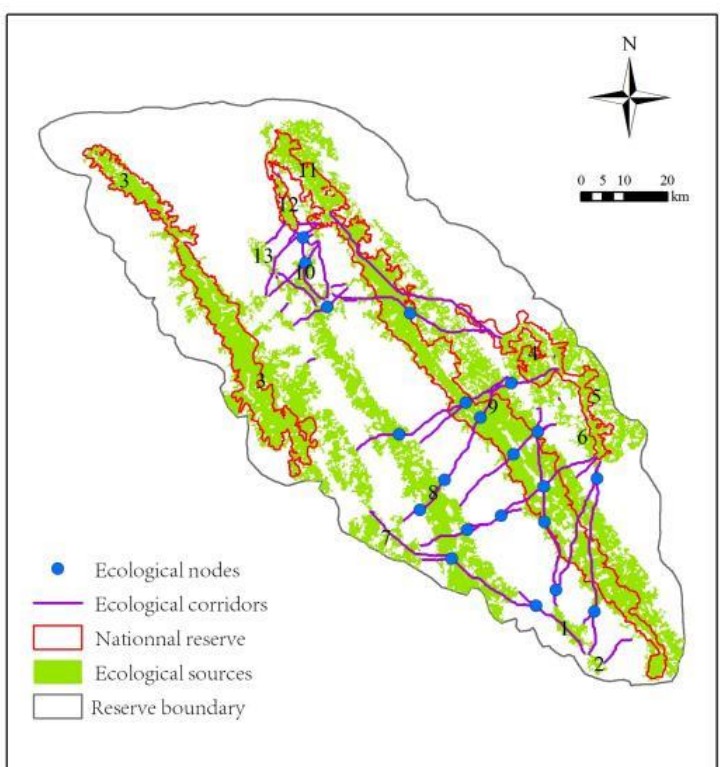

**Figure 4.** Ecological network of the national park.

*3.3. Ecological Network Construction*

3.3.1. MCR Based on the Analysis of Potential Ecological Corridor Extraction

The integrated resistance surface (e) of the Ailao–Wuliang Mountains National Park was constructed based on the vector data of land tenure factor (a), geographic factor (b), vegetation factor (c) and anthropogenic factor (d) in the study area of Figure 3. It can be observed from Figure 3e that the high resistance values in the integrated resistance surface are concentrated in the city centers and villages, which are located outside the scope of the national park and are disturbed by human activities, and their corresponding resistance values are correspondingly higher, while the low resistance values are mostly distributed within the scope of the national park and in the ecological source areas mostly in woodlands. Based on the landscape resistance surface, the MCR model was used to calculate the minimum cumulative resistance value between each ecological source site, and a total of 77 potential ecological corridors were generated, with a total length of 3589 km, to construct the potential ecological corridor of the Ailao–Wuliang Mountains National Park (Figure 3f). As we can see from the figure, the potential ecological corridors in the national park show a denser network with a more uniform spatial distribution, connecting large ecological patches in the park, with more complex corridors among source locations and higher connectivity.

3.3.2. Important Analysis of Ecological Corridors

We numbered the 13 ecological source sites and calculated the interaction strength between different ecological source locations in the study area by the gravity model (Table 4), and the stronger the interaction force between ecological source locations was, the more meaningful the construction of intersource corridors. Based on the study, the gravity threshold was set to 700, and 48 corridors were selected as important corridors with a total length of 865 km (Figure 4). According to Table 4, the interaction strength between source site 9 and source site 11 was the largest at 647,208,185, indicating the strongest spatial association between the two, and the less resistance species encounter when migrating and spreading between the two patches, the more beneficial for regional ecological conservation.

We should therefore strengthen ecological corridor protection between source sites 9 and 11, maintain connectivity of both patches and avoid destruction due to expansion of regional landmasses. Sources 4 and 5 have stronger interaction strengths with sources 5 and 6, indicating that the connectivity between source 4 and source 5 is stronger. The species need to overcome less resistance when propagating movement through the corridor, and the possibility of material and energy exchange is higher, so the ecological corridor between sources 4, 5, and 6 can be established to increase the possibility for species migration between sources 3, 4, 5 and 6 and expand the species' range of activities. Additionally, ecological corridors built between source sites 8 and 9 may link species exchange between source locations 3, 4, 5, and 5. On this basis, the migration and dispersal channels of species between sources 3, 4, 5 and 6 were established, compensating for the high resistance and habitat fragmentation of migration among the sources. For example, the G values between patches 2 and 12 and between patches 2 and 13 were split into 33 and 34, which were distant and poorly connected. The possibility of western black-crowned gibbon dispersion between them was small, and the cost of building ecological corridors was steep if necessary. Accordingly, to improve the possibility of species migration, 25 footstones were established at the convergence point (ecological corridor intersection) and bridge zone where the least expensive paths were selected, and the presence of footstones may compensate for the lack of connectivity of the corridors.

**Table 4.** Level of interaction of ecological corridors.

| Code | 1 | 2 | 3 | 4 | 5 | 6 | 7 | 8 | 9 | 10 | 11 | 12 | 13 |
|------|---|---|---|---|---|---|---|---|---|----|----|----|----|
| 1 | 0 | 313,473 | 742 | 788 | 1244 | 861 | 1122 | 22,956 | 88,786 | 151 | 168 | 77 | 79 |
| 2 | | 0 | 267 | 263 | 413 | 279 | 334 | 2419 | 12,682 | 64 | 68 | 33 | 34 |
| 3 | | | 0 | 1507 | 1002 | 678 | 2,176,830 | 204,027 | 11,100 | 209,177 | 5797 | 3804 | 34,679 |
| 4 | | | | 0 | 27,378,454 | 5,488,676 | 894 | 3684 | 5,649,277 | 769 | 765 | 319 | 307 |
| 5 | | | | | 0 | 18,938,739 | 738 | 2071 | 60,051 | 545 | 555 | 242 | 237 |
| 6 | | | | | | 0 | 528 | 1648 | 137,802 | 279 | 289 | 129 | 126 |
| 7 | | | | | | | 0 | 127,908 | 3154 | 341 | 259 | 140 | 174 |
| 8 | | | | | | | | 0 | 255,194 | 10,341,338 | 9047 | 3752 | 7646 |
| 9 | | | | | | | | | 0 | 122,793 | 647,208,185 | 367,09 | 7410 |
| 10 | | | | | | | | | | 0 | 246,249 | 32,476 | 2,019,026 |
| 11 | | | | | | | | | | | 0 | 1,145,712 | 7539 |
| 12 | | | | | | | | | | | | 0 | 11,611 |
| 13 | | | | | | | | | | | | | |

### 3.3.3. Analysis of Ecological Corridor Construction in Nature Reserves

The ecological network constructed by important ecological corridors is more suitable for areas with fewer villages and farmland. Considering that, in reality, the proposed ecological network will occupy a large amount of land, causing land pressure and aggravating the human–land conflict affecting socioeconomic development. The ecological corridors were constructed by using the patches in the three protected areas as ecological sources, and a total of three ecological corridors, corridors 3-11, 3-12 and 3-9, were generated in the protected areas of the Ailao–Wuliang Mountains. The buffer area resistance cost accumulation values were calculated for each 200 m buffer on each side of the three ecological corridors (Table 5). The table shows that the lowest resistance value for corridors 3-12 is 90.88, which indicates that building this ecological corridor is the least expensive and easiest to achieve. The corridors generated by the Wailing Mountains and Shuangbai Reserve overlap by 11-4, 12-4 and 9-4, respectively, and finally, 3-12-4 can be identified as the optimal ecological corridor for the three reserves (Figure 5), connecting the three nature reserves of the Ailao Mountains, the Wuliang Mountains and the Dinosaur River and included in the scope of the national park.

**Table 5.** Ecological corridor cost table.

| Code | MIN | MAX | MEAN | SUM | |
|------|------|------|------|------|---|
| 3-12 | 0.308381 | 0.750667 | 0.441144 | 90.875575 | √ |
| 3-11 | 0.300714 | 0.785 | 0.481216 | 125.116101 | |
| 3-9 | 0.309476 | 0.814667 | 0.501656 | 129.42734 | |

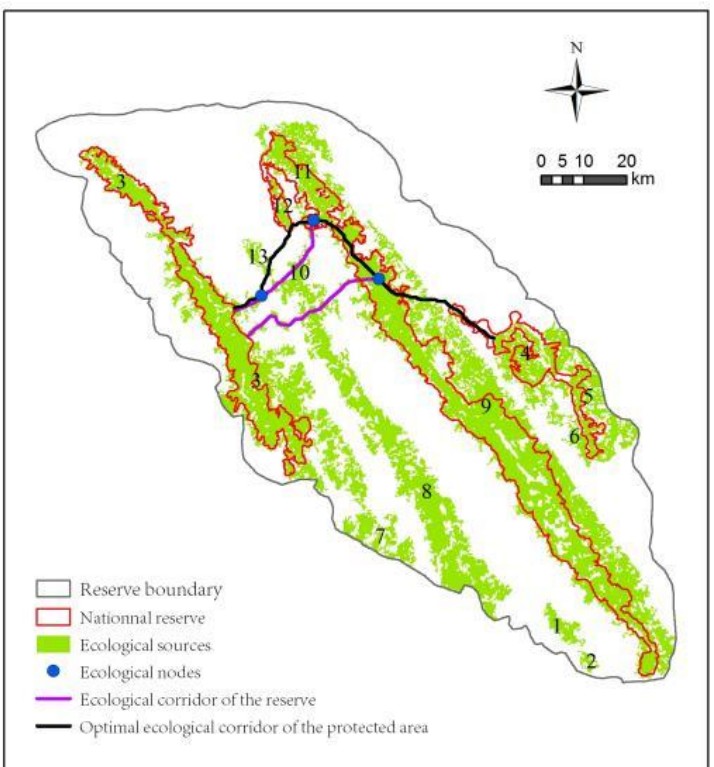

**Figure 5.** Reserve ecological corridors.

3.3.4. Ecological Network Connectivity Evaluation

The structural rationality of potential ecological corridors, important ecological corridors and optimal ecological corridors in protected areas was assessed based on graph theory and network analysis methods (Table 6). The results of the study are summarized in Table 6. Table 6 shows that the $\alpha$ values of potential ecological corridors, important ecological corridors and optimal ecological corridors of protected areas are 1.18, 0.76 and 2, indicating that the optimal ecological corridor of protected areas has the best structural connectivity and better routes for the migration and dispersal of species. The $\beta$ values were 3.08, 2.28, and 1.33, all with $\beta > 1$, which indicated that all were complex structures of ecological networks with high connectivity of ecological corridors. The $\gamma$ values were 1.12, 0.84, and 1.33, where potential ecological corridors and optimal ecological corridors of protected areas had larger $\gamma$, which indicated that their ecological nodes were well connected. The c values were 0.98, 0.94, and 0.97, indicating that the cost values of building both potential ecological corridors, important ecological corridors and optimal ecological corridors in protected areas were higher, and the reason for their higher cost might be interference from anthropogenic activities such as farmland, construction land, and cities in plots between protected areas. Together with the complex geomorphology and fragmented protected areas of the Ailao Mountains and Wuliang Mountains National Park, which lead to the complex structure of ecological corridors, if ecological corridors are constructed in reality, comprehensive consideration is given to the priority of constructing optimal ecological corridors within protected areas.

**Table 6.** Ecological corridor connectivity evaluation table.

| Connectivity Index | Potential Ecological Corridors | important Ecological Corridors | Optimal Ecological Corridor of the Protected Area |
|---|---|---|---|
| $\alpha$ | 1.18 | 0.76 | 2 |
| $\beta$ | 3.08 | 2.28 | 1.33 |
| $\gamma$ | 1.12 | 0.84 | 1.33 |
| c | 0.98 | 0.94 | 0.97 |

## 4. Discussion

### 4.1. Advantages and Challenges of Research Methods Based on MSPA and MCR Models

In this study, an integrated construction method for ecological networks based on the MSPA and MCR was proposed. Compared to the ecological source site identification and ecological network construction of Giant Panda National Park and Shuangzi Mountain National Forest Park using 3S technology and landscape ecology theory [30,31]. MSPA was a widely used method for ecological source site identification, which is simpler and more scientific in distinguishing spatial patterns in the landscape and identifying patches with more suitable conditions as ecological source sites. The combined approach with the MCR model to construct ecological networks has become mature and is commonly used in cities with good results. However, the ecological network constructed by combining these two models has rarely been studied for connected isolated Chinese nature reserves. The ecological source sites in the study area analyzed by the MSPA method in this study were more concentrated and have better landscape connectivity, which is very favorable for the construction and optimization of the ecological network [20]. The ecological corridors and ecological networks constructed by the MCR model are more reasonable and improve the connectivity and integrity of the proposed Ailaoshan–Wuliang Mountain National Park. However, the selection of ecological source sites was the key to improving landscape connectivity and building ecological networks, and in the process of ecological source site selection, source sites can be identified from a multi-indicator integrated evaluation method of ecosystem functional importance [55], biodiversity [56], and species distribution, which can consider the functions, processes, and patterns of ecological source sites in an integrated manner and may lead to one-sided results if ecological source sites were identified from a single level [55]. However, for most areas, species movement and distribution data are often difficult to obtain [57]. Therefore, how to introduce species distribution into the construction of ecological networks using other models remains to be investigated.

### 4.2. Proposed Construction of the Ecological Network of Ailaoshan-Wuliang Mountain National Park

Due to increasing human activities, nature reserves are becoming "islands", which are mostly unable to protect species populations and natural ecological processes in the long term [58]; therefore, there is a need to integrate the reserves into a larger spatial scale to enhance the ecological connectivity among the reserves [59]. China first proposed the establishment of a national park system in 2013 [60], which is comparable to the national parks established internationally, such as Yellowstone National Park in the United States [61], Canadian national parks [62] and the national parks now established in China, which belong to a concentrated contiguous area and were relatively large, but most of the nature reserves in China are insular [63], varying in size and fragmented in distribution, with little connectivity and integrity [64]. By constructing ecological corridors and ecological networks, connectivity among nature reserves can be enhanced and constitute large national parks. Ailaoshan–Wuliangshan National Park has intact wet evergreen broad-leaved forest ecosystems, and both are home to a large number of flagship species of western black-crowned gibbons [65] surrounded by a large amount of remnant forest. History suggests that the Ailaoshan–Wuliang Mountains may have strong connectivity,

providing the possibility for national parks to construct potential ecological networks and ecological corridors.

Influenced by human activities, the construction of a reasonable ecological network requires highlighting the role of anthropogenic disturbance factors in the resistance surface. As shown in Figure 3f, the proposed ecological corridor of Ailaoshan–Wuliangshan Mountain National Park is more evenly distributed. Corridors with high resistance values are located mainly in town centers near villages and roads. In contrast, ecological corridors within protected areas and far from human activities have lower resistance values and can better connect ecological source sites. The resistance surface is usually constructed using unidimensional indicators such as slope, elevation, land use type, roads and human activities [66,67], and the weight values are set by the expert scoring method [22]. In this study, the resistance surface was constructed from multiple dimensions of land tenure, vegetation type, topography and human interference, and the integrated resistance surface model established by using the expert scoring method to set higher weight values for human interference factors, including town center, village, road and land use type, achieves better results in this empirical evidence.

### 4.3. The Impact of Building Ecological Networks on Surrounding Land

The ecological network formed by the proposed 77 ecological corridors is an ideal ecological network, and the types of land they pass through include agricultural land, natural forests, and construction land. If all of them are to be realized, they will occupy a large amount of land and aggravate the conflict between people and the land around them. Although protected areas in Yunnan, China, are located in remote mountainous areas, the surrounding population is large and dependent on land resources [68]. The three protected areas will be set up as ecological source sites, and an optimal ecological corridor will be screened out by calculating the cumulative value of resistance costs in the buffer zone and incorporated into the land area of the national park to implement strict protection management and maximize coordination between the national park and local community residents for conservation and development. In response to the problem of constructing ecological corridors that occupy the surrounding residents' farmland, local special resources can be developed through community participation in co-management and the establishment of ecological compensation mechanisms [69,70]. By encouraging community participation, the conservation and development of national parks coexist. In most cases, large corridors do not preclude reasonable human use of their resources [58]. Combining conservation with the benefits of social, economic, and peripheral development allows residents to share the benefits of natural resource conservation [71], which can weaken the negative effects on the economic development of local communities caused by the occupation of land resources due to the establishment of ecological corridors.

### 5. Conclusions

This study attempted to build an ecological network of the proposed Ailao–Wuliang Mountains National Park based on the MSPA and MCR models. First, based on the MSPA method, we can directly identify and quantify the ecological source sites in the study area and provide important data for the building of ecological networks in national parks. The landscape importance among ecological source locations was further analyzed scientifically using the more scientific Conefor 2.6 software. The integrated resistance surface is generated by the four dimensions of geography, human activities, vegetation, and land tenure, where the pattern of human activities plays a key role in generating the integrated resistance area, and vegetation and land permanence play a major role in the integrated resistant surface. Potential ecological corridors of the national park are generated using the MCR model, and important ecological corridors in the study area were judged based on the assessment of the gravity model. Form an ecological network of the study area, install footstones to optimize the ecological network, and finally screen an ecological corridor to communicate and link protected areas to form a comprehensive ecological

system. The construction of an ecological network of national parks solves the problem of insularity of nature reserves, improves the connectivity and integrity of reserves, solves the real problem of socioeconomic development of the surrounding area caused by the construction of ecological networks by screening the optimal ecological corridors, relieves the pressure on humans and land, and provides the possibility of species migration in the reality of protected areas. The methods based on the MSPA and MCR models are generally applicable to the construction of ecological networks of national parks with multiple nature reserves in isolation, and the results of the study can maximize the conservation of species habitats and biodiversity for the proposed Ailaoshan–Wuliangshan Mountain National Park.

**Author Contributions:** Conceptualization, C.Y. and X.H.; methodology, C.Y. and X.H.; software, X.H.; validation, C.Y.; formal analysis, C.Y. and X.H.; investigation, C.Y., X.L. and X.H.; resources, H.G.; data curation, C.Y. and X.H.; writing—original draft preparation, C.Y. and X.C.; writing—review and editing, Y.W. and X.H.; visualization, C.Y.; supervision, Y.W. and X.H.; project administration, H.G. and Y.W.; funding acquisition, Y.W. All authors have read and agreed to the published version of the manuscript.

**Funding:** This research was funded by the National Natural Science Foundation of China [No. 42061004].

**Data Availability Statement:** Not applicable.

**Acknowledgments:** The authors gratefully acknowledge the support of the funding.

**Conflicts of Interest:** The authors declare no conflict of interest.

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
