# Peer review of "Ecological Network Construction of a National Park Based on MSPA and MCR Models: An Example of the Proposed National Parks of “Ailaoshan-Wuliangshan” in China"

_land, doi:10.3390/land11111913_

Round 1

Reviewer 1 Report

This study combined the MSPA model and the MCP model and constructed an ecological network in Ailaoshan-Wuliangshan National Park. The structure and the pattern of the established ecological network were then analyzed in detail. The authors have done a lot of work and focused on an issue with practical significance for ecological protection. However, there still exist many problems in the manuscript. Below are the details.

1. There are massive language errors, non-standard wordings, and non-standard formats in the manuscript, including a large number of incomplete English sentences, wrong capitalization of initial letters, non-standard wording, and inconsistent font size. These problems exist in the full text, from the abstract to the conclusion section, which greatly affects the quality of this manuscript and hinders reading. It is strongly recommended that the author carefully and thoroughly check the language errors in the manuscript and professionally polished it.

2. The author did not state clearly the significance of the work in this paper. Why do they establish ecological networks in national parks? It is well understood to build an ecological network in cities (referred to by the authors in Line 77) because cities are covered by large areas of built-up land with scattered ecological source sites; by establishing ecological networks, ecological source sites in cities can be well identified, better connected, and better protected. However, in my understanding, national parks are wholely well protected and restricted for construction, and there might be no fragmented ecological habitats. So what is the significance of establishing an ecological network in a national park?

3. The review is not comprehensive and convincing. The author mentioned in the introduction that "Two main research methods exist for building ecological networks...", and these two methods exactly to be the two that have been combined in the article. The word "main" sounds very subjective and vague. Are the authors sure that there are no other methods for constructing ecological networks, and none of them are worth mentioning?

4. The lack of discussion and comparison of methods in this paper makes the innovation of this study unconvincing. It seems that the innovation is the combination of MSPA and MCP to construct ecological networks in national park areas (this combined method has only been used in cities before). So have any existing research constructed ecological networks in national parks? Which method did they use? What is the advantage of combining MSPA and MCP in a national park?

5. The title is misleading and confusing. The authors actually did not carry out the "construction" of the national park but only constructed an ecological network for the national park.

6. The authors highlight the combination of MSPA and MCA, but they did not clearly point out the combination principle in the manuscript.

7. The study area is not clear. The authors mention "parks" in the title, but the term "park" is used in the area introduction. Are the two mountains located in one national park or two national parks?

8. Sections of Discussions and conclusions are strange. The discussion section strangely listed several points with strange logic, and the conclusion section does not point out core conclusions. They need to be rewritten.

Author Response

Dear Professors,

Thank you very much for processing our manuscript (land-1959679) entitled " Ecological network construction of a national park based on MSPA and MCR models: An example of the proposed national parks of “Ailaoshan-Wuliangshan” in China", by Caihong Yang,Xiaoyuan Huang,Junhui Guo,Yanxia Wang,Xiaona Li and Xinyuan Cui. We highly appreciate your comments and revision suggestions. All the comments from the three reviewers are constructive and valuable.I will respond to each of the suggestions given by the experts.

Reviewer 1:

  1. In response to the large number of linguistic errors, irregular wording, and irregular formatting in the manuscript given by the experts, including a large number of incomplete English sentences, incorrect capitalization of initial letters, irregular wording, and inconsistent font size, the manuscript has been carefully corrected and sent to the AJE retouching agency recommended by the journal for retouching.
  2. This paper has explained the significance of constructing an ecological network of national parks in lines 59-63 of the article. Due to the fragmented distribution and fragmentation of nature reserves in China, their variable sizes and small reserves, and the large amount of remaining forests, villages, towns and agricultural land distributed, ecosystem integrity is blocked and integrity and connectivity are not robust. Also, the importance of protected area connectivity is illustrated in lines 64-68.
  3. Other methods for constructing ecological networks proposed by experts have been modified and added. A comparison is made between the method of constructing ecological source sites (lines76-82) and the method of constructing ecological corridors (lines99-102).
  4. The methodological innovations and comparisons proposed by the experts have been explained in Article 3. What are the methods proposed by the experts for the study of ecological networks in existing national parks are explained in lines 112-122 of this paper. Existing studies on the construction of ecological networks of national parks in China are less, and this paper makes comparisons from the studies of birds in Twin Mountain National Park, Giant Panda National Park and Urban National Park.Combined with MSPA is to find ecological source sites by identifying the spatial pattern of forest landscape. The concentrated and contiguous ecological source areas have the most complete ecosystems and the richest biodiversity, and are the main distribution areas of biological habitats. the large core types in MSPA analysis are suitable as ecological source areas. At the same time, these areas are isolated and not well connected, and need to be connected. The corridors should be established in such a way that the ecological cost of the biological flow is minimized. That is, a minimum cost analysis using the MCA method can be achieved.
  5. The title has been corrected to address the confusing nature of the title as suggested by the experts.
  6. The principle of combining MSPA and MCR methods proposed by experts, first: the ecological source site is determined based on MSPA, the larger the ecological source site, the higher the integrity and the richer the biodiversity, but actually it is not the larger the better.Ecological source sites are determined based on area: one is based on the area range of the study area, and the other is the spatial location of the source site distribution (areas with less distribution of large patches will have higher importance of the ecological function of the large patches, and the area of ecological source sites in the area should be appropriately reduced to the area limit).In this paper, 13 ecological source sites were selected, and an area greater than 33km2was chosen as ecological source sites, but to ensure connectivity this paper has 4 source sites with an area smaller than required.Secondly, the more networks constructed is not the better; the networks constructed are too complex and take up a lot of non-forest land, which is difficult to achieve. The little network connectivity is not strong enough to achieve the ecological protection connectivity and integrity effect.
  7. In response to the experts' suggestion that the study area is unclear whether the two mountains are located in one national park or two national parks, which should be the result of inaccurate wording in this paper, changes have been made. Mount Liaojun and Mount Wuliang belong to one national park.
  8. The discussion and conclusion of this article have been rewritten according to the requirements and the content of the article.

Reviewer2:

  1. In response to experts who pointed out many editorial problems in the manuscript, especially in the introduction and data analysis sections (lack of spaces between words, lack of spaces after periods, etc.). Changes have been made.
  2. Some statements in the chapters in which the expert proposes the study area are followed by citations (e.g., statements in lines 98-99, 99-102, 104-105, 110-111), putting the references found as far as possible have been added. (e.g. line 152 references [34], line 162 citing references [35,36], line 164 citing references [37], line 167 citing references [38]).
  3. The issue that the scientific names of the species should be italicized as pointed out by the experts has been revised.

Reviewer3:

  1. Punctuation issues raised by experts have been revised
  2. determine the weight values according to the expert scoring method in references [45,46].
  3. It has been revised according to the issues raised by experts to avoid duplication.
  4. Has been revised in response to the expert's question about adding perspectives to the conclusions.
  5. The ecological network constructed in this paper can improve the connectivity among protected areas, maximize species exchange and protect biodiversity. It can be used as a reference for other nature reserves and national parks in China to improve their integrity and connectivity.

 Our point-to-point responses and corrections are displayed with red fonts in the following text. We have made major and careful revisions closely following the comments. Thus, the quality and clarity of our manuscript are improved. We sincerely hope our responses are satisfactory.

If you have any question, please let us know.

Yours sincerely,

Caihong Yang

Name: Xiaoyuan Huang

  • mail: hxy21cn@swfu.edu.cn

Reviewer 2 Report

First of all I would like to congratulate the authors for their work. 

Then I would like to point out that the manuscript has many editorial problems especially in the introduction and data analysis parts (lack of space between words, lack of space after the period, etc.).

I suggest that some statements in the chapter describing the study area be followed by citations (statements in rows 98-99, 99-102, 104-105, 110-111). Also some statements could be more specific e.g. row 36 which of the Chinese regions are biodiversity hotspots. 

The scientific names of the species should be italicised. 

Author Response

(The authors gave the same response as above.)

Reviewer 3 Report

It is a very good piece of work on biological conservation through the construction of ecological network (corridor). The paper is well written and the results seems good. I just have few remarks on:

1. the type of MS Word version used because the words are stucks with punctuation or not. I have underlined some.

2. The methodology can be improved. I wonder myself What exactly help you in giving each weight i.e 0.5, 0.3 etc. In other words, how do you find or calculate those weight values? (Lines 193-199, page 5).

3. There are some repetitions that could be avoided. I have selected them in track mode.

4. Lines 470-473 (page 15). The authors can add perspectives in their conclusion.

5. Two questions for my own understading: would you think that what you did in this paper  is more beneficial for conservation and will help protecting efficiencies those flagship species? What lessons can  national conservation policies learn from your work?

Author Response

(The authors gave the same response as above.)

Round 2

Reviewer 1 Report

All my concerns have been well addressed.